# Multi-Transmotion: Pre-trained Model for Human Motion Prediction

**Yang Gao,   Po-Chien Luan,   Alexandre Alahi**
Visual Intelligence for Transportation (VITA) laboratory
EPFL, Switzerland
`{firstname.lastname}@epfl.ch`

**Abstract:** The ability of intelligent systems to predict human behaviors is crucial, particularly in fields such as autonomous vehicle navigation and social robotics. However, the complexity of human motion have prevented the development of a standardized dataset for human motion prediction, thereby hindering the establishment of pre-trained models. In this paper, we address these limitations by integrating multiple datasets, encompassing both trajectory and 3D pose keypoints, to propose a pre-trained model for human motion prediction. We merge seven distinct datasets across varying modalities and standardize their formats. To facilitate multimodal pre-training, we introduce Multi-Transmotion, an innovative transformer-based model designed for cross-modality pre-training. Additionally, we present a novel masking strategy to capture rich representations. Our methodology demonstrates competitive performance across various datasets on several downstream tasks, including trajectory prediction in the NBA and JTA datasets, as well as pose prediction in the AMASS and 3DPW datasets. The code is publicly available: https://github.com/vita-epfl/multi-transmotion.

**Keywords:** Human motion prediction, Trajectory prediction, Pose prediction, Multimodal pre-trained model, Multitask pre-trained model

## 1 Introduction

The research community has witnessed substantial advancements through the adoption of pre-trained models. In natural language processing (NLP), large language models (LLMs) have demonstrated remarkable interdisciplinary proficiency, excelling across a variety of downstream tasks [1, 2]. In contrast, pre-trained models in computer vision (CV) typically exhibit a greater degree of task specificity, which can be attributed to the multimodal nature of visual data [3, 4]. This specificity typically results in reduced efficiency when compared to NLP models. Nevertheless, recent developments in multitask pre-training, exemplified by MultiMAE [5] and 4M [6], have shown promising capabilities in transferring knowledge across a diverse array of CV tasks. However, this critical gap remains in the field of human motion prediction. Unlike fields such as NLP and CV, human motion incorporates rich representations and manifests through diverse modalities, including keypoints, trajectories, and bounding boxes, each reflecting different aspects of human movement. Despite this complexity, there currently exists no multimodal pre-trained model for accurately predicting human motion. Intuitively, human motion cannot be fully expressed by a single modalities. Thus, we argue that each modality can benefit from the others by integrating multiple modalities into the models. Consequently, the development of a multitask pre-trained model is imperative for this domain.

Three principal challenges must be overcome to effectively pre-train a model for human motion prediction. First, the field lacks a comprehensive, large-scale dataset that encompasses various modalities of human motion. Second, a versatile framework is required to handle these diverse modalities, in contrast to previous approaches that typically addressed each modality in isolation. Third, the model must be robust when confronted with incomplete or noisy input data. To address

8th Conference on Robot Learning (CoRL 2024), Munich, Germany.

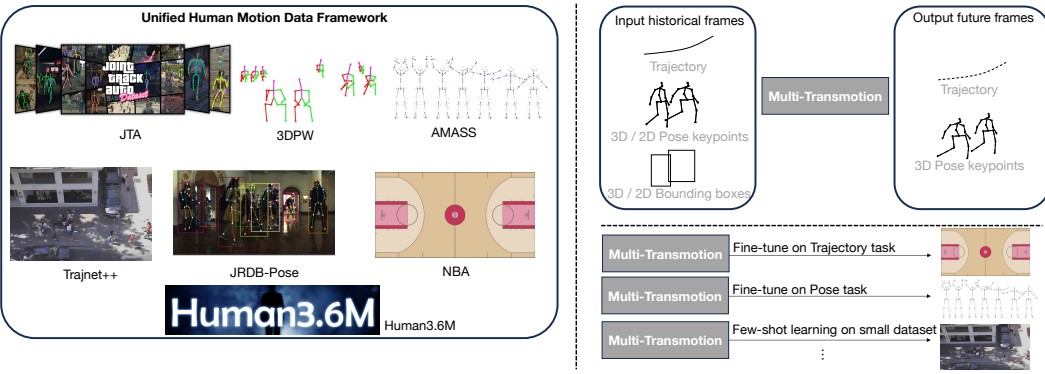

Figure 1: **Overview.** We propose a unified human data framework by standardizing the data format and frame settings. Based on this framework, we introduce a pre-trained transformer model with specialized masking techniques, validating its effectiveness and flexibility across different scenarios.

these challenges, this paper proposes a novel approach that integrates multiple datasets, develops a flexible network architecture, and demonstrates its effectiveness in handling noisy data.

Due to the absence of a large-scale multimodal human motion prediction dataset, we have undertaken the task of merging several existing datasets, namely Joint Track Auto (JTA) [7], Trajnet++ [8], JRDB-Pose [9], NBA [10], Human3.6M [11], AMASS [12] and 3DPW [13]. Each dataset was originally created with different data formats and frame settings. To streamline the training process, we have unified the data framework. This framework standardizes the observation and prediction horizons, as well as the frame rates, ensuring consistency across the merged datasets. This unified framework enables more efficient and effective pre-training of the model, addressing the complexities of multimodal human motion prediction. For the model design, we implement a tokenization strategy that maintains spatial-temporal information across all modalities by applying modality-specific linear projection layers to convert coordinates into hidden dimensions. To further enhance the model's robustness and adaptability, we employ up-sampling padding, sampling masks, and a bi-directional temporal encoder. The up-sampling padding facilitates easy fine-tuning across different frame rates, while the sampling masks simulate various frame rates by masking tokens with different chunk sizes.

Figure 1 shows an overview of our work. By unifying the datasets, we are able to pre-train a transformer-based model, Multi-Transmotion, that can predict future trajectories and informative 3D pose keypoints. The flexibility of our model architecture and data framework allows for easy fine-tuning to specific tasks with varying frame settings. Our model achieved competitive results on both tasks, demonstrating the effectiveness of our pre-training techniques, as shown by our ablation studies on few-shot learning and robustness.

We summarize the main contributions as follows:

- **Dataset**: We create a unified human motion data framework by merging seven datasets with standardized settings. Additionally, this framework is flexible, allowing for the seamless addition of more datasets or adjustments to frame settings.

- **Method**: We propose Multi-Transmotion, a pre-trained transformer-based model that flexibly adapts to different frame settings, demonstrating strong robustness and efficiency. This model outperforms previous models across several datasets.

## 2 Related Work

**Human trajectory prediction** involves forecasting the future positions and movements of individuals based on their past and current trajectories [14]. Data-driven methods have demonstrated remarkable efficacy in human trajectory prediction [15, 16, 17, 18]. Numerous studies have explored social interactions to enhance prediction accuracy, such as social pooling [8, 19, 20], graphs [21, 22, 23,

24, 25, 26], and attentions [27]. Diverse architectures have been explored including recurrent neural networks (RNNs) [15, 23, 8], generative adversarial networks (GANs) [19, 28, 29, 30], conditional variational autoencoders (CVAEs) [31, 32], diffusion models [33, 34, 35], and LLMs [36]. Moreover, integrating transformer architectures with positional encoding has become a useful tool for capturing long-range dependencies and is widely used in trajectory prediction tasks [37, 38, 39, 40]. From a dataset perspective, trajdata [41] provides a unified interface for handling trajectory and map data. Recent studies have also explored leveraging 2D body pose as visual cues for trajectory prediction in image space [42], with an emphasis on the utility of an individual agent's 3D body pose for trajectory prediction [43, 44, 45]. Thus, a foundational model for this task must adeptly leverage available visual cues, including pose information, to produce accurate predictions.

**Human pose prediction** involves predicting the future coordinates of pose keypoints. The pose keypoints can be extracted with monocular [46] or stereo [47] cameras. When predicting the poses, recurrent neural networks have traditionally been prominent [48], capitalizing on their ability to capture temporal dependencies in sequential data. Subsequent advancements introduced feed-forward networks as alternatives [49], while graph convolutional networks (GCNs) were proposed to better capture the spatial dependencies of body poses [50]. Noteworthy innovations include separating temporal and spatial convolution blocks [51] and introducing trainable adjacency matrices [52, 53]. Diffusion [54] has also shown strong performance by repairing and refining pose. Transformer-based approaches have also gained traction for modeling human motion [55, 56], showcasing significant improvements with spatio-temporal modules [50]. Accordingly, the pre-trained model proposed in this work incorporates attention mechanisms to capture both temporal and spatial pose representations.

**Masked prediction** is a technique used to learn robust representations by reconstructing masked data. BERT [57] uses masked prediction in natural language processing by replacing random words in a sentence with a special mask token and training the model to predict these masked words from the surrounding context. In computer vision, Masked Autoencoder (MAE) [58] applies a similar concept by masking random patches of an input image and training an autoencoder to reconstruct the missing patches, thereby learning meaningful image representations. The masking strategy has also been extended to trajectory prediction. Forecast-MAE [59] and Traj-MAE [59] both leverage the concept of masked autoencoders to enhance the tasks of motion prediction and trajectory prediction, respectively. Traj-MAE employs diverse masking strategies to pre-train trajectory and map encoders, capturing social and temporal information while utilizing a continual pre-training framework to mitigate catastrophic forgetting. Similarly, Forecast-MAE utilizes a novel masking strategy that considers the interconnections between agents' trajectories and road networks, demonstrating competitive performance and substantial improvements over previous self-supervised methods in motion prediction.

## 3 Method

### 3.1 Unified Human Motion Data Framework

As previously mentioned, one of the primary challenges in the human motion domain is the variety of datasets. Specifically, different datasets vary in terms of horizon, frame rates, data formats, and pose joint configurations. To address the dataset challenges, we propose a unified human motion dataset framework with standardized settings. We generate data sequences under uniform conditions in a consistent format, specifically 2 seconds of observation and 4 seconds of prediction at 5 frames per second (fps). Since we are handling human motion prediction tasks, all annotations are recorded as coordinates. Regarding pose data, different datasets have varying joint ID configurations. Our unified joint ID configuration is provided in Appendix A.1. As depicted in Table 1, the dataset framework comprises 7 datasets featuring diverse combinations of modalities. By combining synthetic and real-world data, this framework encompasses over **2 million samples** for trajectories and over **1 million samples** for 3D pose keypoints. To our best knowledge, it is the largest data framework for human motion prediction currently available. Furthermore, this dataset framework allows for the flexible addition of new data and adjustments to the horizon with different frame rates.

Table 1: **Approximate number of annotations of the training split of our unified human motion data framework.** The default setting predicts 4 seconds into the future with 2 seconds of observation at 5 fps, except for TrajNet++ [8]*, where the frame setting is fixed to 2.5 fps.

| Dataset | R(eal)/S(ynt.) | Traj | 3D BB | 2D BB | 3D Pose | 2D Pose | Scene |
|---|---|---|---|---|---|---|---|
| NBA SportVU [10] | R | 430k | / | / | / | / | No |
| Trajnet++ [8] (w/ ETH/UCY [60, 61])* | R+S | 250k | / | / | / | / | No |
| JRDB-Pose [9] | R | 284k | 284k | 284k | / | 94k | Yes |
| JTA [7] | S | 764k | 764k | 764k | 764k | 764k | Yes |
| Human3.6M [11] | R | 256k | 256k | 256k | 256k | 256k | Yes |
| AMASS [12] | R | 314k | 314k | / | 314k | / | No |
| 3DPW [13] | R | 4k | 4k | / | 4k | / | Yes |
| SUM | R+S | 2302k | 1622k | 1304k | 1338k | 1114k | |

## 3.2 Multi-Transmotion - A Pre-trained Transformer-based Model for Human Motion Prediction

Pre-trained models have demonstrated remarkable effectiveness across diverse domains, from language tasks [62] to image tasks [63, 3]. However, a significant gap remains in the development of pre-trained models specifically designed for human motion prediction. This gap stems from the absence of a model architecture capable of managing multimodal motion prediction with the flexibility to accommodate different frame settings and keypoint configurations. To bridge this gap, we proposed Multi-Transmotion (Figure 2), a multimodal pre-trained model for multi-task motion prediction, designed to seamlessly integrate all available visual cues while adapting to **varying horizons**, **frame rates**, and **pose keypoints**. To achieve this, we devised dynamic spatial-temporal mask, sampling mask, and bi-directional temporal encoding strategies, enhancing the model's robustness and adaptability to different frame rates and observation horizons. Detailed math reasoning was shown in Appendix A.6.

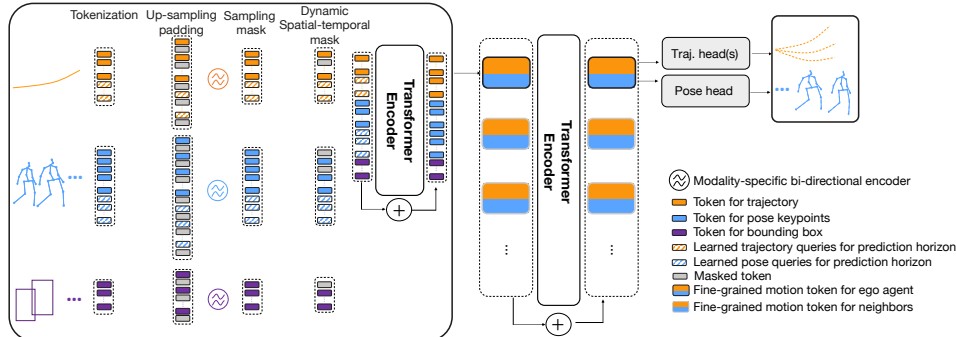

Figure 2: **Multi-Transmotion: A transformer-based model that learns cross-modality representations and social interactions.** Sampling mask and bi-directional encoders make the model flexible to different frame settings, while dynamic spatial-temporal mask make pre-training more efficient and robust.

**Tokenization.** To retain the spatial-temporal information across all modalities, we apply modality-specific linear projection layers to tokenize the coordinates into the hidden dimension of the transformer. Specifically, we project 3D x-y-z coordinates for trajectories and 3D bounding boxes/pose keypoints into the hidden dimension. Likewise, the pixel coordinates of 2D bounding boxes/pose keypoints in image space are projected into the hidden dimension. The learned queries are padded and initialized as future motion tokens. These tokens are concatenated with historical motion tokens, after which masking strategies are applied. The combined tokens are then processed by the transformer.

**Up-sampling padding, sampling mask, and bi-directional encoder.** To ensure the model can be easily fine-tuned with different frame rates, we apply up-sampling padding to the valid tokens during pre-training, simulating the maximum fps for the model. The maximum fps is set to 50, as it can be easily downsampled to commonly used fps settings for trajectory prediction tasks (2.5 fps and

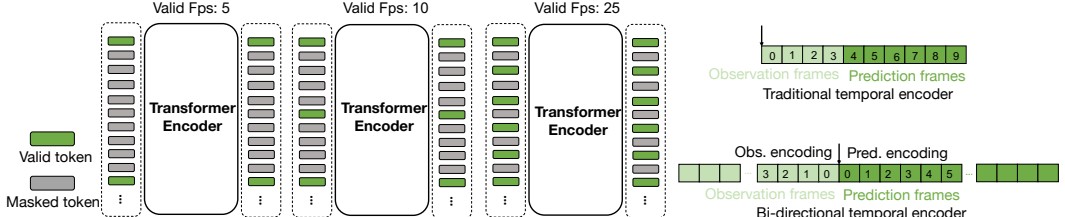

Figure 3: **Sampling mask and bi-directional encoder.**

5 fps) [44, 8, 33] and pose prediction tasks (25 fps and 50 fps) [54, 50]. As illustrated on the left of Figure 3, the sampling mask allows the model to mask different chunk sizes, simulating various frame rates. The right of Figure 3 shows the difference between the traditional temporal encoder and our bi-directional temporal encoder. Rather than starting the encoding from the first observation, we begin from the last observation and encode observations and predictions separately, facilitating seamless adaptation to different observation and prediction horizons during the fine-tuning process.

**Dynamic Spatial-temporal mask.** After applying the bi-directional encoder and sampling mask, each valid token includes spatial-temporal information, allowing the model to learn detailed positional representations. To make the model more robust, we introduce a dynamic spatial-temporal mask, which randomly masks out a dynamic spatial-ratio ($0 < r_s < 1$) of the total number of 3D/2D pose keypoints for all frames. Meanwhile, we apply a temporal-ratio ($r_t = 0.1$) for trajectory and 3D/2D bounding box modalities, randomly masking some frames of data. This dynamic spatial-temporal mask strategy has made pre-training more effective compared to related works [44, 5], as demonstrated in our ablation study shown in Appendix A.3.

## 4 Experiments

### 4.1 Datasets

We exclude Trajnet++ [8] and use the training split of the other 6 datasets to to develop a pre-trained model, enabling us to perform few-shot learning on Trajnet++ [8]. For the evaluation of the trajectory prediction task, we employ the NBA [10] and JTA [7], as they represent the largest trajectory-only dataset and the largest dataset with 3D pose, respectively. To assess performance on pose prediction tasks, we use the widely adopted large-scale AMASS [12] and the smaller-scale 3DPW [13], to thoroughly evaluate performance in both indoor and outdoor scenarios. Additionally, we also show the model's application on robot navigation task in Appendix A.2 and the robustness against imperfect poses in Appendix A.4.

Regarding experimental settings, we follow the pioneering works [33, 44, 54] with the configurations below:

- NBA [10]: Predict 4 seconds of trajectory given 2 seconds of past trajectory at 5 fps.

- JTA [7]: Predict 4.8 seconds of trajectory given 3.6 seconds of past trajectory at 2.5 fps.

- AMASS [12]: Predict 1 second of pose keypoints given 2 seconds of past pose at 25 fps.

- 3DPW [13]: Predict 1 second of pose keypoints given 2 seconds of past pose at 25 fps.

### 4.2 Metrics

For evaluation, we use Average Displacement Error (ADE) and Final Displacement Error (FDE) for the deterministic trajectory prediction task on JTA [7], and MinADE$_K$/MinFDE$_K$ as best-of-k ADE/FDE for stochastic trajectory prediction on NBA [10]. Mean Per Joint Position Error (MPJPE) is used for the pose prediction tasks on AMASS [12] and 3DPW [13].

Table 2: **Quantitative results on NBA [10].** $MinADE_{20}/MinFDE_{20}$ (meters) are reported.

| Models | Venue | $MinADE_{20}$ | $MinFDE_{20}$) |
|---|---|---|---|
| Social-LSTM [15] | CVPR 16 | 1.65 | 2.98 |
| Social-GAN [19] | CVPR 18 | 1.59 | 2.41 |
| STGAT [21] | ICCV 19 | 1.40 | 2.18 |
| Social-STGCNN [26] | CVPR 20 | 1.53 | 2.26 |
| Trajectron++ [23] | ECCV 20 | 1.15 | 1.57 |
| NPSN [64] | CVPR 22 | 1.31 | 1.79 |
| GroupNet [24] | CVPR 22 | 0.96 | 1.30 |
| MID [65] | CVPR 22 | 0.96 | 1.27 |
| Leapfrog [33] | CVPR 23 | 0.81 | 1.10 |
| Social-Transmotion [44] | ICLR 24 | 0.78 | 1.01 |
| Multi-Transmotion | | **0.75** | **0.97** |

## 4.3 Results

**Trajectory prediction on NBA [10].** We selected the large-scale NBA [10] to evaluate our model on the pure trajectory prediction task, as this dataset does not provide visual modalities. To ensure a fair comparison with pioneering works, the test split remained hidden from the model during the pre-training process. We used the same test data provided by [33] for evaluation. Table 2 presents a numerical comparison between our method and several strong baselines. Our approach, which involves pre-training followed by fine-tuning, consistently outperformed all prior baselines. This result underscores the effectiveness of our proposed architecture, which integrates innovative masking and encoding techniques, establishing its value as a pre-trained model.

Figure 4 illustrates the qualitative results of our model, showing that the predicted trajectory closely aligns with the ground truth. This provides further evidence of the model's strong performance in highly interactive sports scenarios.

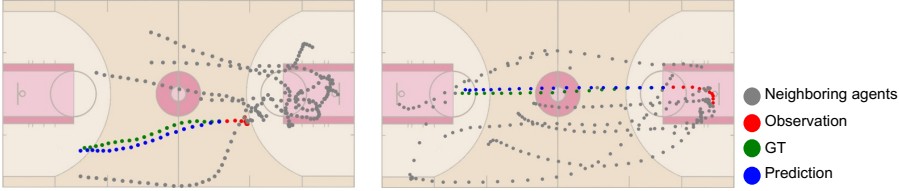

Figure 4: **Qualitative results of on NBA [10].** The red, blue, and green dots represent the observed historical frames, predicted future frames, and ground truth, respectively. All neighboring players are shown in grey.

**Trajectory prediction on JTA [7].** To demonstrate the effectiveness of 3D pose in augmenting trajectory prediction, we evaluate the model using pure trajectory (T) and trajectory combined with 3D pose (T+3D P) as input modalities on the JTA [7]. Since 3D pose is the most informative visual cue, we fine-tune our model with trajectory and 3D pose, then compare it with Social-Transmotion [44] trained on the same modalities to ensure fairness. As shown in Table 3, our model effectively leverages knowledge from 3D poses to enhance trajectory prediction. Furthermore, our model demonstrates superior performance in utilizing pose knowledge compared to the previous multimodal trajectory prediction model [44], which we attribute to our pre-training approach and novel masking strategy. Despite the different frame settings from pre-training, the model's performance remains robust through fine-tuning, demonstrating the flexibility of our pre-trained model.

To further examine how the model leverages 3D pose information, we visualize the predictions of our model using Trajectory-only and Trajectory + 3D pose. Figure 5 shows the qualitative results of our model. It is evident from this figure that the last frame of the pose provides valuable information

Table 3: **Quantitative results on JTA [7].** Social-Transmotion [44] and Multi-Transmotion are trained on Trajectory and 3D Pose modalities since they can leverage pose. ('T' and 'P' abbreviate Trajectory, and Pose keypoints)

| Models | Venue | Input modality at inference | ADE | FDE |
|--------|-------|------------------------------|-----|-----|
| Vanilla-LSTM [15] | CVPR 16 | T | 1.44 | 3.25 |
| Directional-LSTM [8] | T-ITS 21 | T | 1.37 | 3.06 |
| Social-LSTM [15] | CVPR 16 | T | 1.21 | 2.54 |
| Autobots [39] | ICLR 22 | T | 1.20 | 2.70 |
| Trajectron++ [23] | ECCV 20 | T | 1.18 | 2.53 |
| EqMotion [40] | CVPR 23 | T | 1.13 | 2.39 |
| Social-Transmotion [44] | ICLR 24 | T | 0.99 | 2.00 |
| Social-Transmotion [44] | ICLR 24 | T + 3D P | 0.94 | 1.94 |
| Multi-Transmotion | | T | 0.97 | 1.97 |
| Multi-Transmotion | | T + 3D P | **0.91** | **1.89** |

about walking direction and body rotation, enabling the model to predict a more accurate future trajectory when pose information is available.

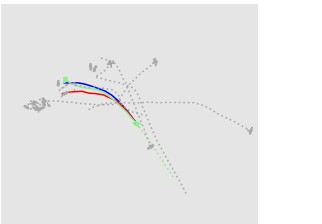 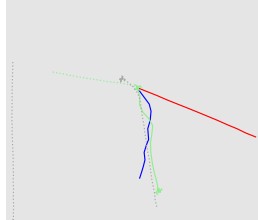 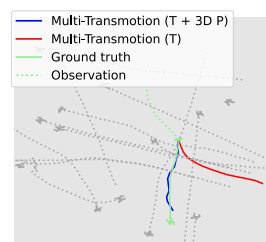

Figure 5: **Qualitative results on dataset.** This visualization shows how our model can leverage 3D pose information to augment trajectory prediction. The red trajectory denotes the prediction without using pose knowledge, and the blue trajectory denotes the prediction with help of leverage 3D pose. The green trajectory denotes the ground truth.

**Pose prediction on AMASS [12] and 3DPW [13].** For the pose prediction task, we utilize the AMASS [12] and 3DPW [13] to thoroughly evaluate our model's performance in indoor and outdoor scenarios. Following previous works [54], we use the same test split and data preprocessing to normalize the poses. Table 4 presents the qualitative results on the AMASS and 3DPW, showing that our model consistently achieves the best or second-best performance across the entire prediction horizon.

Table 4: **Quantitative results on AMASS [12] and 3DPW [13].** Numbers are reported by MPJPE in millimeter at specific predicted time-step. **Best** numbers are in bold and second-best numbers are underlined.

| Model | AMASS [12] | | | | 3DPW [13] | | | |
|-------|-----------|--------|--------|---------|-----------|--------|--------|---------|
| | 160 ms | 400 ms | 720 ms | 1000 ms | 160 ms | 400 ms | 720 ms | 1000 ms |
| ConvSeq2Seq [49] (CVPR 18) | 36.9 | 67.6 | 87 | 93.5 | 32.9 | 58.8 | 77 | 87.8 |
| LTD-10-10 [66] (ICCV 19) | **19.3** | 44.6 | 75.9 | 91.2 | **22** | 46.2 | 69.1 | 81.1 |
| LTD-10-25 [66] (ICCV 19) | 20.7 | 45.3 | 65.7 | 75.2 | 23.2 | 46.6 | 65.8 | 75.5 |
| HRI [50] (ECCV 20) | 20.7 | 42 | 58.6 | 67.2 | 22.8 | **45** | **62.9** | **72.5** |
| ST-Trans [56] (Ra-L 24) | 21.3 | 42.5 | **58.3** | **66.6** | 24.5 | 47.4 | 64.6 | 73.8 |
| Multi-Transmotion | **19.3** | **41.4** | 58.6 | 66.9 | 22.5 | 45.6 | 64.2 | 73.7 |

Figure 6 presents the qualitative results of our approach, with the actions of sitting, walking, and standing shown in separate rows. Benefiting from pre-training on the large-scale motion dataset, the walking scenario (second row) is handled effectively by our model. In the sitting scenario (first row), the prediction quality declines in the last few frames as the agent begins to stand up, a challenging transition even for humans to predict. Overall, our model demonstrates competitive performance on the pose prediction task, evaluated intensively across various scenarios in AMASS [12].

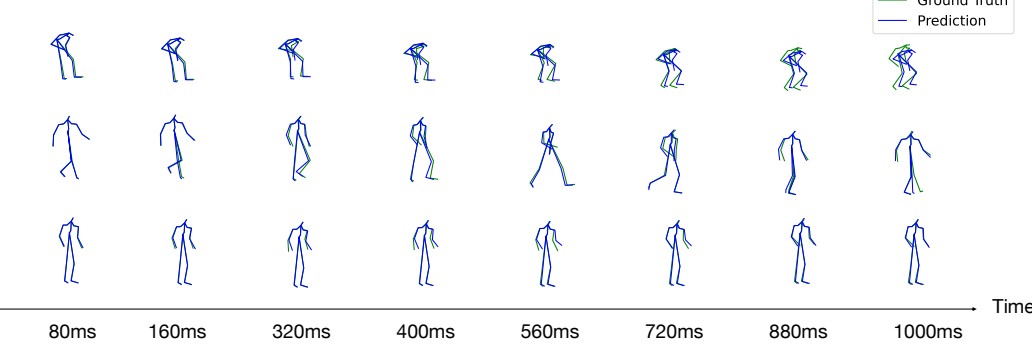

Figure 6: **Qualitative results on AMASS [12]**, with sitting, walking, and standing shown by row.

**Few-shot learning on small data under new frame settings**    To further explore the potential of our model on small datasets, we conduct few-shot learning on a maximum of 1k samples and evaluate on approximately 4k samples from the real-world part of Trajnet++ [8]. We fine-tuned our model to adapt to the Trajnet++ [8] frame settings and compared it with a transformer-based baseline model (Autobots[39]) and an LSTM-based baseline model (Social-LSTM [15]). Figure 7 shows that Social-LSTM converges faster than Autobots, as transformer networks are typically very data-hungry. However, our Multi-Transmotion model outperforms Social-LSTM, benefiting from the pre-trained knowledge.

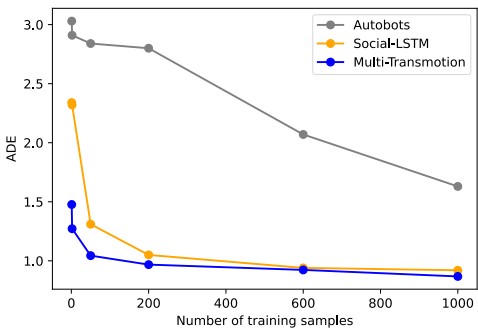

Figure 7: **Few-shot learning curve**

Table 5: **Comparison of Pre-trained model and Specific model.** Specific model indicates the model is trained from scratch on one dataset. $MinADE_{20}/MinFDE_{20}$ are used for NBA [10], ADE/FDE are used on Trajnet++ [8], and final MPJPE is used on AMASS [12] and 3DPW [13].

| Dataset | Pre-trained model | Specific model |
|---|---|---|
| NBA [10] | 0.75/0.97 | 0.77/0.98 |
| Trajnet++ [8] | 0.54/1.13 | 0.57/1.22 |
| AMASS [12] | 66.91 | 69.58 |
| 3DPW [13] | 73.74 | 76.77 |

**Pre-trained model vs. Specific model**    To explore the generalization capabilities of models trained on different datasets, we conduct an ablation study to see if the model benefits from pre-trained knowledge. We observe the performance differences between a pre-trained model trained with multiple datasets and specific models trained on individual datasets. Table 5 demonstrates that the pre-trained model consistently outperforms the specific models. This indicates that pre-training on large-scale data significantly enhances the generalization ability of the model for both human trajectory prediction and pose prediction.

## 5   Conclusion and Limitations

In this work, we introduced a unified human motion data framework and Multi-Transmotion, the first pre-trained transformer-based model in the human motion prediction domain. By employing several novel masking strategies, our model can fine-tune on different frame settings and achieve competitive results on human motion prediction, underscoring its practical value. Additionally, the masking strategies enhance efficiency without sacrificing robustness against noisy input.

As for limitations and future work, incorporating additional modalities, such as contextual images and human intentions, can further empower the model and bring it closer to being a foundational model for predicting human motion. We leave this for future exploration.

## 6 Acknowledgement

The authors would like to thank Saeed Saadatnejad, Lan Feng, Reyhaneh Hosseininejad, and Zimin Xia for their valuable feedback. This work was supported by Sportradar[1] (Yang's Ph.D.) and Honda R&D Co., Ltd.

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

# A Appendix

## A.1 Unified Human Motion Data Framework

Table 6 presents the detailed joint configurations and the mapping between our unified IDs and the original datasets.

Table 6: **Joint ID Mapping**

| Unified ID | Unified joints | JTA Old ID | H36M Old ID | JRDB Old ID | AMASS Old ID | 3DPW Old ID |
|---|---|---|---|---|---|---|
| 0 | pelvis | 15 | 0 | 8 | 0 | 0 |
| 1 | right_hip | 16 | 1 | 10 | 2 | 2 |
| 2 | right_knee | 17 | 2 | 13 | 5 | 5 |
| 3 | right_ankle | 18 | 3 | 15 | 8 | 8 |
| 4 | right_foot_arch | nan | 4 | nan | 11 | 11 |
| 5 | right_toes | nan | 5 | nan | nan | nan |
| 6 | left_hip | 19 | 6 | 11 | 1 | 1 |
| 7 | left_knee | 20 | 7 | 14 | 4 | 4 |
| 8 | left_ankle | 21 | 8 | 16 | 7 | 7 |
| 9 | left_foot_arch | nan | 9 | nan | 10 | 10 |
| 10 | left_toes | nan | 10 | nan | nan | nan |
| 11 | spine (H36M) | nan | 12 | nan | nan | nan |
| 12 | thorax/chest | nan | 13 | nan | nan | nan |
| 13 | neck | 2 | 14 | 4 | 12 | 12 |
| 14 | head_center | 1 | 15 | nan | 15 | 15 |
| 15 | left_shoulder | 8 | 17 | 5 | 16 | 16 |
| 16 | left_elbow | 9 | 18 | 7 | 18 | 18 |
| 17 | left_wrist | 10 | 19 | nan | 20 | 20 |
| 18 | left_outer_thigh | nan | 21 | nan | nan | nan |
| 19 | left_hand | nan | 22 | 12 | nan | 22 |
| 20 | right_shoulder | 4 | 25 | 3 | 17 | 17 |
| 21 | right_elbow | 5 | 26 | 6 | 19 | 19 |
| 22 | right_wrist | 6 | 27 | nan | 21 | 21 |
| 23 | r_outer_thigh | nan | 29 | nan | nan | nan |
| 24 | right_hand | nan | 30 | 9 | nan | 23 |
| 25 | head_top | 0 | nan | 0 | nan | nan |
| 26 | right_clavicle | 3 | nan | nan | 14 | 14 |
| 27 | left_clavicle | 7 | nan | nan | 13 | 13 |
| 28 | spine 0 (JTA) | 11 | nan | nan | nan | nan |
| 29 | spine 1 (JTA) | 12 | nan | nan | nan | nan |
| 30 | spine 2 (JTA) | 13 | nan | nan | nan | nan |
| 31 | spine 3 (JTA) | 14 | nan | nan | nan | nan |
| 32 | right_eye (JRDB) | nan | nan | 1 | nan | nan |
| 33 | left_eye (JRDB) | nan | nan | 2 | nan | nan |
| 34 | spine 1 (AMASS/3DPW) | nan | nan | nan | 3 | 3 |
| 35 | spine 2 (AMASS/3DPW) | nan | nan | nan | 6 | 6 |
| 36 | spine 3 (AMASS/3DPW) | nan | nan | nan | 9 | 9 |
| 37 | nose | nan | nan | nan | nan | nan |
| 38 | forehead | nan | nan | nan | nan | nan |

## A.2 Application in Robot Navigation

To further explore the potential application of our model in robotic tasks, we integrated our predictor into a robotic navigation system.

We applied the CrowdNav [67] simulator to generate pedestrian trajectories with social interactions. Consequently, there were approximately 800 training samples and 200 test samples. The social force

model [68] was used as the navigator, as it allowed us to easily incorporate the predicted trajectories by adding extra repulsive forces and validate the effectiveness of our predictor. As Table 7 shows both the completion time and the collision rate were reduced after incorporating our predictor, which highlighted the effectiveness of our model in robot navigation scenarios. Additionally, we found that the improvement in completion time could reach up to 14% when increasing the social forces.

Table 7: **Application of our model in robot navigation task**

|  | **Completion time in seconds (gain)** | **Collision rate (gain)** |
|---|---|---|
| robot navigation w/o our predictor | 16.46 | 1.93% |
| robot navigation w/ our predictor | **16.20** (+2%) | **0.39%** (+80%) |

## A.3 Comparison Between Different Masking Strategies

Masking has been shown to efficiently improve the robustness and generalization ability in transformer architectures [58, 44]. In this ablation study, we pre-train three smaller models on JTA [7] to quickly examine how different masking implementations affect pre-training in human motion prediction. This ablation study is implemented on the same Tesla V100 GPU device. Table 8 shows that while modality-mask and meta-mask yield strong performance, they come with high computational costs. Conversely, the fixed spatial-temporal mask used in [5] improves computational efficiency by consistently dropping a fixed number of tokens but leads to the lowest robustness. Our implementation employs a dynamic spatial-temporal mask that randomly drops tokens, maintaining high computational speed without sacrificing robustness during pre-training.

Table 8: **Effect of different masking strategy.**

|  | **Modality-mask and Meta-mask [44]** | **Fixed Spatial-temporal mask [5]** | **Dynamic Spatial-temporal mask (ours)** |
|---|---|---|---|
| Computational cost | | | |
| Spatial cost | 6864 MB | 3128 MB | 6142 MB |
| Temporal cost | 11.306 ms | 7.631 ms | 7.349 ms |
| Performance on JTA [7] | | | |
| T | 1.11/2.25 | 1.50/3.13 | 1.00/1.99 |
| T + 3D B + 2D B | 1.04/2.08 | 1.13/2.23 | 0.99/1.98 |
| T + 3D B + 2D B + 3D P + 2D P | 0.96/1.94 | 0.95/1.93 | 0.96/1.94 |

## A.4 Does Imperfect Pose Input Still Augment Trajectory Prediction?

In the real world, it is challenging to capture accurate 3D pose keypoint annotations due to occlusion or sensor noise. Therefore, it is crucial to assess whether the model remains reliable when faced with incomplete or noisy pose input. To better understand the robustness in such scenarios, we also compare with a specific model trained without any masking strategy. Table 9 presents the performance comparison, showing that the masking strategy significantly enhances the model's robustness. Specifically, it reduces performance degradation from 46.7% to 11.0% when dealing with noisy 3D pose keypoints.

Table 9: **Performance under imperfect pose input.**

| **Input Modality at inference** | **Multi-Transmotion** ADE / FDE (degradation% ↓) | **Specifc model w/o masking** ADE / FDE (degradation% ↓) |
|---|---|---|
| T + 100% 3D P | 0.91 / 1.89 | 0.92 / 1.90 |
| T + 50% 3D P | 0.92 / 1.90 (1.1% / 0.5%) | 0.99 / 1.98 (7.6% / 4.2%) |
| T + 10% 3D P | 0.95 / 1.94 (4.4% / 2.6%) | 1.17 / 2.31 (27.2% / 21.6%) |
| T + 3D P w/ Gaussian Noise (std=25) | 0.97 / 1.98 (6.6% / 4.8%) | 1.16 / 2.31 (26.1% / 21.6%) |
| T + 3D P w/ Gaussian Noise (std=50) | 1.01 / 2.05 (11.0% / 8.5%) | 1.35 / 2.68 (46.7% / 41.1%) |

In addition to simulating noisy pose input, it is more realistic to handle estimated poses from existing pose estimators. In this study, we use the estimated 3D poses generated by an off-the-shelf pose estimator [69] on some clips from the JRDB-Pose [9]. Table 10 shows that our model can also generalize to pseudo 3D poses, improving trajectory prediction compared to using only trajectory data. Although the improvement is modest, it still demonstrates the model's effectiveness in a realistic scenario.

Table 10: **Performance with using estimated Psudo 3D pose**.

| Model | ADE | FDE |
|---|---|---|
| Multi-Transmotion (T) | 0.13 | 0.20 |
| Multi-Transmotion (T + Pseudo 3D P) | 0.11 | 0.18 |

### A.5    Implementation Details

The dual-transformer architecture has 4 heads in each transformer. The first transformer, used to learn multimodal features, has 6 layers, while the second transformer, focused on social interaction, has 4 layers. For the pre-training, we use the Adam optimizer [70] with a learning rate starting at $1 \times 10^{-4}$, which decays by a factor of $0.1$ after 80% of the 60 total epochs are completed. The pre-training is conducted on 8 NVIDIA A100 GPUs, each with 80GB of memory. L2 loss is applied to both the trajectory prediction and pose prediction tasks. No supervision is applied to masked keypoints (e.g., missing joints) and the model is trained to predict the available pose keypoints annotated in the data.

### A.6    Math Reasoning for Method

In this work, all terms regarding trajectory, 3D/2D bounding box, and 3D/2D pose refer to the 3D/2D coordinates of that modality. E.g., we use 2D x-y coordinate input/output for the bird-eye-view trajectory and 3D x-y-z coordinate input/output for the 3D pose. Specifically, we denote the sequential Trajectory, 3D/2D bounding box and 3D/2D local pose of agent $i$ as $x_i^{Traj}$, $x_i^{3dB}$, $x_i^{2dB}$, $x_i^{3dP}$, and $x_i^{2dP}$, respectively. The observed time-steps and predicted time-steps are defined as $t = 1, ..., T_{obs}$ and $t = T_{obs} + 1, ..., T_{pred}$.

Considering a sample with $N$ pedestrians, the **model input** is $X = [X_1, X_2, ..., X_N]$, where $X_i = \{x_i^c, c \in \{Traj, 3dB, 2dB, 3dP, 2dP\}\}$ depending on the availability of different modalities. The tensor $x_i^c$ has a shape of $(T_{obs}, e^c, f^c)$, where $e^c$ represents the number of elements in a specific cue (e.g., the number of 3D pose keypoints) and $f^c$ denotes the number of features (e.g., 3 for x-y-z dimension of 3D pose keypoints) for each element.

Lastly, the **model output** is the ego (i=1) pedestrian's motion $Y = Y_1$, where $Y_1 = \{Y_{Traj}^k, Y_P\}$ containing $k$ future possible sequential trajectories and one future sequential deterministic 3D pose keypoints.

During the tokenization process, we use modality-specific MLPs to encode each input modality, resulting in the projected hidden dimension $H_i^c = MLP^c(\mathbf{x_i^c})$. Next, we apply Up-sampling Padding and a modality-specific Bi-directional encoder to generate tensors with high-frequency positional information, i.e., $H_i^c = U(H_i^c) + B^c$, where $B^c$ is the Bi-directional encoder applying to modality $c$ and $U$ denotes Up-sampling padding. After this step, each token from $H_i^c$ represents a specific element of one modality at a specific time step (e.g., a token could contain information about the 3D neck keypoint at the first observed frame).

The token length of each modality can be calculated as $L_c = e_c * t_c$, where $e_c$ is the number of elements in a modality and $t_c$ is the number of frames considered for that modality. During the masking process, we first use the Sampling Mask to mask out specific chunk sizes based on the data configurations, allowing the model to simulate different frame settings. Then, the Dynamic Spatial-temporal mask randomly drops a dynamic spatial-ratio ($0 < r_s < 1$) of the total number of 3D/2D pose tokens for all frames. Meanwhile, we apply a temporal-ratio ($r_t = 0.1$) for tokens

related to trajectory and 3D/2D bounding box modalities, as each of these modalities contains only one element, randomly masking some frames of data. After applying the masking, the token length for the first transformer ($L_1$) is given by the following formula:

$$L_1 = L_{Traj} + L_{3dB} + L_{2dB} + L_{3dP}(1 - r_d) + L_{2dP}(1 - r_d).$$

As the second transformer learns the fine-grained motion interaction between pedestrians, the token length for the second transformer ($L_2$) is calculated as follows:

$$L_2 = N(L_{Traj} + L_{3dP}(1 - r_d)),$$

where $N$ is the number of pedestrians in the scene. After passing through the transformers, the fine-grained tensors are denoted as $FH_i^c$, where $c$ represents the output modalities of the model. I.e., $c \in \{Traj, 3dP\}$

Finally, we use modality-specific MLP heads to project the multi-modal x-y trajectories and deterministic x-y-z 3D pose keypoints for the ego (i=1) pedestrian:

$$Y_{Traj}^k = MLP_{Traj}^k(FH_1^T), \quad Y_P = MLP_P(FH_1^{3dP}).$$

