# OpenReview forum: "Multi-Transmotion: Pre-trained Model for Human Motion Prediction"
_robot-learning.org/CoRL/2024/Conference — CoRL 2024_

### Official Review · Reviewer_ePRe · 2024-07-18
**I think this paper is not prepared well**

**Originality:** 2
**Technical Quality:** 2
**Clarity Of Presentation:** 2
**Potential Impact:** 2
**Recommendation:** 2
**Confidence:** 4

**Review:**

This paper summarizes its contribution from two perspectives: Dataset and Method. However, in the introduction, the authors state, "Due to the absence of a large-scale multimodal human motion prediction dataset, we have undertaken the task of merging several existing datasets," and all the experiments are conducted on single datasets. It seems this paper trains a multi-channel transformer with a combined dataset and tests the performance on single datasets.

The proposed framework looks similar to [42] in the paper, while the key difference is sampling methods without a clear explanation of the framework design, and the comparison is not complete with other methods, where [42] reported the result with T+ 3D P is not the same as the paper reported. which is not show the advance in specific tasks. Because other scenarios such as T + 2D P, T + 2D BB, T + 3D P + 3D BB, and T + 3D P + 2D P + 3D BB + 2D BB is missing.

The proposed framework is hard to understand without any mathematical reasoning, I cannot understand the exact input and output of the proposed methods and the exact problem the proposed methods want to address. If it is only one pretrained framework can deal with multiple datasets, the novelty is not enough.

**Quality Of The Limitations Section:**

2

**Questions For Rebuttal:**

1. How the multi-transmotion works? It is hard to understand the novelty of the proposed framework without any mathematical reasoning.
2. The comparison between methods did not report correctly and completely.
3. The sampling is only to align with different fps, the novelty is limited.

**Robotics Focus:**

3

**Summary Of Paper:**

The paper "Multi-Transmotion: Pre-trained Model for Human Motion Prediction" proposes a new transformer-based model to predict human motion more accurately by leveraging a novel integration of multiple datasets that include trajectory and 3D pose keypoints. This integration aims to address the lack of a standardized, large-scale multimodal dataset for human motion prediction. The authors developed a unified framework to standardize these datasets, which allowed them to pre-train their model effectively. The model, named Multi-Transmotion, utilizes novel masking strategies to learn rich representations and demonstrates competitive performance on several downstream tasks across different datasets. The study emphasizes the model's robustness and adaptability in handling various modalities and datasets, showing promising results in trajectory prediction for NBA and JTA datasets and pose prediction in the AMASS and 3DPW datasets.

**Summary Of Recommendation:**

I am currently leaning towards rejection of this paper with lack of method explaination and comprehansive comparision.

---

### Official Review · Reviewer_6LnL · 2024-07-18
**Nice unifying dataset and interesting architecture**

**Originality:** 5
**Technical Quality:** 5
**Clarity Of Presentation:** 4
**Potential Impact:** 3
**Recommendation:** 3
**Confidence:** 5

**Review:**

The unified human motion dataset is a very nice contribution to the community as evaluating new methods on standard datasets remains tedious and time consuming for researchers. In addition, the proposed transformer-based method that utilizes multiple modalities is a welcoming and novel addition. Overall, the paper provides a good contribution and is well-written.

To make the paper better, this reviewer recommends the following:
- Explain how the impact of other humans can be accounted for in the proposed architecture, or, clearly state the lack thereof in the Limitations section.
- Explain how other important modalities can be added (this should be possible with the proposed architecture): Human motion prediction is nuanced, this work is missing some important features commonly found in other work. These include, but not limited to: the environment (e.g., occupancy maps), latent vectors encoding the influence of other agents (e.g., [1]), robot camera observation
- Explain how few-shot learning works: Since the proposed method only has transformer encoders, it is not directly clear how the model can do few-shot learning. The authors should explain this clearly.
- Explain the training loss for reproducibility.
 -There are quite a few design choices, e.g. bidirectional positional encoding, Dynamic Spatial-Temporal Mask. It would be great to see ablation studies on these as practitioners may want to know how important these are.

Here are some minor issues:
- It is very strange to call a projection to a continuous vector “tokenization” as it usually refers to converting strings to s sequence of integers. Perhaps projection is a better term. Also, how is the projection done? It is by a multiplying with a matrix? Pass through a MLP? Cross-attend with a learned query vector?
- It would be nice to state how many parameters the model has.

**Quality Of The Limitations Section:**

1

**Questions For Rebuttal:**

How is the few-shot learning done? Can you show a scaling curve, i.e., the performance as a function of number of demonstration examples (shots)

**Robotics Focus:**

3

**Summary Of Paper:**

This paper tackles the absence of a unified framework and pre-trained models for multimodal human motion prediction. It proposes a two-pronged approach: constructing a large-scale dataset by integrating and standardizing seven existing datasets with trajectory, pose and bounding box modalities. In addition, it proposes Multi-Transmotion, a transformer-based model pre-trained on this unified dataset. The authors evaluate the model on established benchmarks for trajectory prediction (NBA, JTA) and 3D pose prediction (AMASS, 3DPW), reporting competitive results on standard metrics. Additionally, they explore the model's performance in few-shot learning scenarios.

**Summary Of Recommendation:**

This work is of high quality and is quite novel. The evaluation is sufficient to demonstrate that the proposed method advances the state of the art in human motion prediction

---

### Official Review · Reviewer_4Ldu · 2024-07-31
**Review of Multi-Transmotion: Pre-trained Model for Human Motion Prediction**

**Originality:** 2
**Technical Quality:** 2
**Clarity Of Presentation:** 3
**Potential Impact:** 2
**Recommendation:** 3
**Confidence:** 3

**Review:**

**Pros:**

The paper is well written and easy to understand. The overall aim of the paper is also interesting and the problem being tackled is definitely quite relevant. The experimental results do seem to show that the approach works well enough in most cases.  The overall idea of dynamic spatio-temporal masking seems interesting compared to modality specific masking and it seems to provide some improvements over it.

**Cons:**

I am not sure if simply combining a lot of existing datasets will be sufficient to train a large pre-trained model. This is especially because these existing datasets often represent very different activities being performed and hence the dynamics of human motion can vary. For instance, the existing paper combines NBA Sport VU with Human 3.6M (which is concerned with a diverse set of human motions). It is unclear if there is a huge positive transfer between these datasets which can be quite different. The other problem is that the joint mapping between some of these datasets is not consistent (and sometimes missing) and hence its unclear if naively combining all of this data is going to give consistent benefits.

*Limited novelty:* There is also limited technical novelty compared to prior works such as Social-Transmotion, traj-MAE which already proposed modality masking etc.

*Results:* While the overall results seem positive they definitely don’t have a huge difference when compared to baselines. For example the difference between proposed approach and Social-Transmotion is only 0.02-0.03 (in terms of displacement error). Moreover, even for single-dataset training scenarios (i.e. comparing the benefits of pre-training vs training on a single downstream dataset) the difference is around 0.02-0.03 (displacement error for trajectory prediction). This seems like a huge of computational effort is spent for pre-training to get very little benefits. Similarly, even for few-shot learning examples very old LSTM based approaches (Social-LSTM) seem competitive with proposed approach (difference of only 0.01 in terms of displacement error). Overall, while the results are mostly positive, it’s unclear if they are practical since they require much larger compute and do comprehensively beat baselines. Do the authors have any thoughts on why this happens? Also, maybe the authors should consider expanding upon this more in their paper.

*Relevance to robotics:* I think one big thing I see is the relevance of the paper for the robotics community. While human motion prediction is an important problem, its unclear if the target CoRL attendees will be the right audience for this paper.

**Quality Of The Limitations Section:**

1

**Questions For Rebuttal:**

please see above.

**Robotics Focus:**

2

**Summary Of Paper:**

The paper focuses on the problem of human motion prediction and aims to develop a large scale model for human motion prediction. The hope is that such a large scale pre-trained model can provide a good prior for training motion prediction model on smaller datasets. To train such a large pretrained model the paper collects existing datasets and pre-trains on them. Since these datasets have different sampling frequencies some special care has to be taken when using them together for pre-training. This is achieved by sampling to a fixed high-frequency rate (50 FPS) and then using extra masked tokens to represent intermediate missing data points.

**Summary Of Recommendation:**

The paper is interesting and focuses on an important topic. However, the effectiveness of merging disparate datasets for high-quality pre-training remains questionable.  While the results are encouraging, they do not present significant advance from previous approaches.

---

### Official Review · Reviewer_CsuV · 2024-08-02
**Multi-Transmotion - Novel transformer training method and architecture applied to human motion prediction. Dataset creation and standardization was crucial.**

**Originality:** 4
**Technical Quality:** 4
**Clarity Of Presentation:** 3
**Potential Impact:** 3
**Recommendation:** 3
**Confidence:** 3

**Review:**

This paper represents a valuable contribution to human motion prediction research. Main contributions are as follows:
* The unified data framework across 7 datasets and standardization that included frame rate, common horizons, coordinate and pixel representations and unified pose joint configuration.
* Model architecture with modality specific tokenization, up-sampling padding and bi-directional temporal encoder.
* Masking during training to randomly hide parts of the motion data which enhances the model's robustness and efficiency.
* State of the art results on several challenging human motion prediction tasks, demonstrating the effectiveness of their approach.
* Few-shot learning on smaller amounts of data to boost performance in narrow domains.

The main weakness of the paper is the lack of empirical evidence of the model's applicability on robotic's tasks like navigation. While it is understandable because it is hard to evaluate on real scenarios, it would be possible to leverage simulation benchmarks (see below).

**Quality Of The Limitations Section:**

2

**Questions For Rebuttal:**

The method and results are very appealing but having result in a robotics task would make the paper even stronger.
* One suggestion would be to do few-shot learning on a sim navigation task and show that the concepts learned can transfer and are applicable to robot navigation. HuNavSim could be a good benchmark: https://arxiv.org/pdf/2305.01303.

**Robotics Focus:**

3

**Summary Of Paper:**

Predicting human motion is crucial for applications like self-driving cars and robots, but existing models often struggle with the complexity and variety of human movement. This paper proposes a new approach:  First, they create a large, unified dataset by combining seven existing datasets that cover different types of motion (walking, sports, etc.).  Second, they develop Multi-Transmotion, a specialized transformer model that can learn from this diverse data. Finally, they use a novel "masking" strategy to train the model, making it more robust to noisy or incomplete data. Their results show that Multi-Transmotion outperforms previous methods and provides a strong foundation for future research in human motion prediction.

**Summary Of Recommendation:**

I recommend a weak accept given the strong performance of the model on human motion prediction benchmarks and novel methods.

---

### Author Rebuttal · Authors · 2024-08-13

Please find the figure and video in the attached 'zip' file, thanks!

---

### Decision · Program_Chairs · 2024-09-04

**Decision:**

Accept

**Comment:**

PRE REBUTTAL:

High level summary of reviews:

Strengths:

- Unified data framework and standardization: The paper presents a valuable contribution by unifying and standardizing seven existing datasets, enabling the training of a large-scale model for human motion prediction.
- Multi-Transmotion model architecture: The novel transformer-based architecture is considered a valuable contribution.
- Masking during training: The masking strategy employed during training enhances the model's robustness and efficiency.
- State-of-the-art results: The model achieves state-of-the-art results on several challenging human motion prediction tasks, demonstrating the effectiveness of the approach.
- Few-shot learning: The model's ability to perform well in few-shot learning scenarios, boosting performance in narrow domains with smaller amounts of data, is seen as a strength.

Weaknesses:

- Limited novelty: Some reviewers found limited technical novelty compared to prior works.
- Questionable dataset merging: The effectiveness of merging disparate datasets for high-quality pre-training is questioned.
- Marginal performance gains: Despite positive results, the improvements over baselines are considered marginal, raising questions about the practical significance of the findings.
- Lack of empirical evidence in robotics tasks: The absence of empirical evidence demonstrating the model's applicability to robotics tasks, such as navigation, is seen as a weakness.
- Limited relevance to CoRL community: The relevance of the paper to the robotics community is questioned, with concerns that CoRL attendees may not be the target audience.
- Lack of clarity and comparison: The paper lacks clarity in explaining the framework design and comparisons with other methods, making it difficult to understand the exact input, output, and addressed problem. Reviewer ePRe also says the framework is hard to understand without mathematical reasoning.

POST REBUTTAL:

While this paper marginally crossed the threshold for acceptance, the rebuttal arguments pertaining to low technical novelty and weak results are somewhat unconvincing. A novel composition of existing approaches is fair game for technical contribution (and some of the most impactful papers are along these lines!), however in such cases the final results are used to justify the engineered algorithm, and that is where I think this work is lacking. The authors argue that beating state of the art performance was not the main focus of the paper, but then the contribution that the resultant algorithm "can adapt" to various input characteristics definitely limits impact (since it's likely that small modifications to SoTA methods would also have this property). However, the authors did provide a thorough rebuttal and addressed many other of the reviewer concerns. While there was minimal reviewer engagement during the rebuttal period, the result is a stronger manuscript that just meets the bar for acceptance.